# Effectiveness of a Multicomponent Training Program on Physical Performance and Muscle Quality in Older Adults: A Quasi-Experimental Study

**DOI:** 10.3390/ijerph20010222

**Published:** 2022-12-23

**Authors:** Noé Labata-Lezaun, Max Canet-Vintró, Carlos López-de-Celis, Jacobo Rodríguez-Sanz, Ramón Aiguadé, Leonor Cuadra-Llopart, Esther Jovell-Fernández, Joan Bosch, Albert Pérez-Bellmunt

**Affiliations:** 1Faculty of Medicine and Health Sciences, Universitat International de Catalunya, 08195 Barcelona, Spain; 2ACTIUM Functional Anatomy Group, 08195 Barcelona, Spain; 3Fundació Institut Universitari per a la Recerca a l’Atenció Primària de Salut Jordi Gol i Gurina (IDIAPJGol), 08007 Barcelona, Spain; 4Nursing and Physiotherapy Department, Universitat de Lleida, 25198 Lleida, Spain; 5Department of Geriatric Medicine, Consorci Sanitari de Terrassa, 08227 Terrassa, Spain; 6Department of Epidemiology, Consorci Sanitari de Terrassa, 08227 Terrassa, Spain

**Keywords:** elderly, muscle quality, physical functional performance, multicomponent training

## Abstract

Aging is associated with a decrease in functional capacity, manifested by a loss of strength, physical performance and muscle quality. Multicomponent training (MCT), characterized by the combination of at least three types of training, could be a good strategy to counteract these changes. To date there are no studies evaluating the effectiveness of MCT in improving both physical performance and muscle quality simultaneously. The aim of this study is to evaluate the changes produced by an MCT program on both physical performance and muscle quality in a population of healthy older adults. Sixteen healthy older adults were recruited to perform a 15-session multicomponent training intervention. Physical performance was assessed by different functional tests, and muscle quality was assessed by tensiomyography and myotonometry. The main results of this study show some improvement in functional tests, but not in muscle quality parameters, except for vastus lateralis stiffness. MCT is able to generate improvements in the physical performance of older adults, but these improvements are not reflected in muscle quality parameters measured by tensiomyography and myotonometry.

## 1. Introduction

During the last decades, the increase in life expectancy has been causing a general aging of the population. In Europe, approximately 20% of the population is over 65 years of age [1]. In addition, aging involves structural [2], metabolic [3], and neuroendocrine changes [4] that influence the musculoskeletal system and ultimately lead to a decrease in physical performance [2]. The World Health Organization (WHO) defined healthy aging as "the process of developing and maintaining functional capacity that enables well-being in old age" [5]. In this sense, emphasis is placed on the preservation of physical performance, which ensures the person’s independence.

For their part, international organizations such as the European Working Group on Sarcopenia in Older People (EWGSOP), emphasize i the importance of the assessment of both strength and physical performance in the diagnosis of sarcopenia in their latest consensus [6]. Moreover, the organization highlights the relevance of assessing not only the quantity, but also the muscle quality (micro and macroscopic aspects of muscle architecture and composition); and admits the current challenge of finding reliable tools capable of assessing it [6]. In recent years, several tools for assessing muscle quality, such as tensiomyography and myotonometry, have been developed. Tensiomyography is a valid and reliable method of assessing muscle quality [7,8]. It assesses stiffness by measuring the radial displacement of the transverse fibers of the muscle belly as a function of the time in which the contraction occurs. Myotonometry is another valid and reliable method of assessing muscle function used to evaluate the viscoelastic properties of tissues [9,10,11,12]. Although both tools are used to assess the intrinsic properties of tissues, their results cannot be exchanged one for the other [13].

Regarding the different strategies to ensure "healthy aging”, physical exercise has been shown to be effective in modulating aging-related changes, preventing muscle atrophy, maintaining cardiorespiratory fitness and maintaining functional independence [14,15,16]. In particular, multicomponent training (MCT) characterized by the combination of at least three types of training such as resistance, aerobic, balance, and/or flexibility training has shown to be effective in improving the physical performance in large sample randomized controlled clinical trials [17] and meta-analyses [18]. In fact, the latest international consensus on exercise in the older adults promotes the implementation of MCT programs as one of the best strategies for improving strength, balance and gait, as well as reducing falls [19].

Although the effectiveness of MCT in improving physical performance has been extensively studied, to date there are no studies evaluating the effectiveness of MCT in improving both physical performance and muscle quality. Thus, the aim of this study is to evaluate the changes produced by an MCT program on both physical performance and muscle quality in a population of healthy older adults.

## 2. Materials and Methods

### 2.1. Study Design

This is a quasi-experimental study. The study protocol was registered in ClinicalTrials.gov with the code NCT05286723.

### 2.2. Participants

Twenty participants were recruited according to their availability to participate through the Associació de Gent Gran Casal Anna Murià (Terrassa). Four participants were finally unable to take part in the intervention, two for COVID reasons, one for having suffered a fall prior to the measurements, and one for musculoskeletal issues.

The study was approved by the Research Ethics Committee of the Universitat Internacional de Catalunya (CBAS-2021-08). All participants signed a written informed consent document before enrolling in the study. Data protection laws were respected according to the Helsinki declaration.

Prior to measurements, the attending geriatricians reviewed the absolute and relative contraindications to participate in the intervention.

The inclusion criteria were (a) persons over 65 years of age, and (b) able to carry out a therapeutic exercise program.

Exclusion criteria were (a) the inability to stand or ambulate in an unassisted manner, (b) previous bone fracture in the last 6 months, (c) uncontrolled symptomatic cardiovascular or respiratory disease, (d) uncontrolled hypertension, (e) current cancer under treatment, and (f) inability to understand the information provided by the researchers.

An attendance percentage of 60% of the training sessions was established in order to be included in the statistical analysis. Drop-out was considered only when the pre-intervention assessment was completed.

### 2.3. Procedure

Once the subjects were contacted, it was confirmed that they met the inclusion/exclusion criteria and they were scheduled for the evaluation. Pre-intervention measurements took place in October 2021. First, anthropometric data were recorded and questionnaires were handed out. Subsequently, muscle quality and physical performance were assessed. At the end of the intervention period, post-intervention measurements were taken in December 2021 following the same structure used previously.

### 2.4. Intervention

The intervention began in November 2021, and took place in the Casal Gent Gran Anna Murià. The intervention consisted of 15 sessions of multicomponent training over a period of 2 months, with a training frequency of two sessions per week. The duration of the sessions was 1 hour. The intervention was carried out by a physiotherapist and a physical trainer. During the sessions, a 15-minute warm-up was performed, which included joint mobility exercises, gait work, active stretching, and balance and coordination exercises. The main part of the training consisted of 30 minutes of strength training, in which pull, push, squat and deadlift patterns were worked using dumbbells of different weights to adjust the workload. The difficulty of the exercises was adapted for each of the participants. Three sets of 8–12 repetitions were performed for each movement, increasing the load to ensure a moderate-high intensity of 7–8 in rate of perceived exertion (RPE). A 90-second rest was taken between each set. The session ended with 15 minutes of unmonitored aerobic exercise, including collaborative games or going up and down stairs. Adherence to the programme was documented in a daily register, and phone calls were made in case the person did not attend the training session.

### 2.5. Variables

The variables measured can be grouped into physical performance variables and muscle quality variables. To assess physical performance, the following tests were used: Short Physical Performance Battery (SPPB), walking test, Five Times Sit-to-Stand Test (5XSST), Timed Up and Go test (TUG), and handgrip strength. The muscle quality parameters assessed were contraction time and maximal radial displacement, measured by tensiomyography, and stiffness as measured by myotonometry.

#### 2.5.1. Short Physical Performance Battery (SPPB)

This is a test battery consisting of three tests. It consists of a balance test, a four meters walking speed test and a test of sitting down and getting up from the chair five times. Each result has a value of 0–4 points which is summed to obtain an overall score of 0–12 points. Test-retest reliability has been shown to be good to excellent (ICC 0.83–0.92), and the inter-rater reliability (ICC 0.91) among acutely admitted older adults was shown to be excellent as well [2].

#### 2.5.2. Meters Walking Test (4WT)

This is a functional test that reflects the average speed in which the subject takes to walk 4 meters. Although it is included in the SPPB battery, its score has a value by itself. The test was performed twice and the one with the shorter time was chosen. Its reliability has been previously studied (ICC = 0.96, 95%CI = 0.94–0.98; SEM = 0.01) [20]. 

#### 2.5.3. Five Times Sit-to-Stand Test (5XSST)

This reflects the time in seconds it takes the person to sit down and stand up five times from a chair, without the help of the arms. Although it is included in the SPPB battery, its score has a value on its own. The test was performed twice and the one with the shorter time was chosen [2].

#### 2.5.4. Timed Up and Go Test (TUG)

This is a functional test that reflects the time in seconds that it takes the person to get up from the chair, with the help of the arms, walk 3 meters, turn around an obstacle, return to the chair and sit down again [6]. The test was performed twice and the one with the shorter time was chosen. Its reliability has been studied previously (ICC = 0.98, 95%CI = 0.93–1.00; SEM = 0.7) [2].

#### 2.5.5. Handgrip Strength

This is a test that reflects the maximal grip strength in kilograms (Kg) using a hand dynamometer [21,22]. The device used was the Jamar® dynamometer. In order to perform the measurements, the subject was placed in a seated position with the arms supported ensuring 90° elbow flexion with the wrists in a neutral position. Both the dominant and non-dominant arms were measured. Three measurements were taken for each hand, with a one-minute rest between measurements. The mean between the three measurements of each hand was calculated and the hand that obtained the best results was chosen. The validity and reliability of this device has been evaluated in previous studies (ICC = 0.98) [2,23]

#### 2.5.6. Maximal Radial Displacement (Dm)

Concentric muscle contraction is produced by the sliding of the actin myofilaments over the myosin myofilaments in the sarcomeres. As a result, a decrease in their longitudinal section is generated, which in turn generates an increase in the cross-sectional area of the muscle. This increase in cross-section is visualized by a displacement in a radial direction, i.e., perpendicular to the muscle belly. Maximal radial displacement of the muscle in millimeters is measured with tensiomyography under an involuntary contraction by electrostimulation [13]. In the present study, the rectus anterior quadriceps and vastus lateralis muscles were measured. To perform the measurement, the patient must be placed in a muscle relaxation position, generally in the decubitus position. Two electrodes (TMG electrodes, TMG-BMC d.o.o., Ljubljana, Slovenia) are placed on the most prominent part of the chosen muscle belly at a distance of 5 centimeters between electrodes. In addition, the Dc-Dc Trans-Tek® transducer (GK40, Panoptik d.o.o., Ljubljana, Slovenia) is placed perpendicularly. Through an electrostimulator device (TMG- BMCd.o.o., Ljubljana, Slovenia) connected to the electrodes, an involuntary contraction of the muscle is generated, causing radial displacement of the sensor and generating a time-displacement curve. The amplitude is progressively increased from 20 to 100 mA by increments of 20 mA until no further increase in Dm is recorded or the maximum point of the stimulator is reached (i.e., 110 mA). Ten seconds of rest was performed between stimuli to minimize the effects of fatigue and post-activation potentiation [24].

#### 2.5.7. Contraction Time (Tc)

This reflects the time measured in milliseconds that the muscle takes to reach 10 to 90% of the maximal radial displacement (Dm) [24]. In the present study, the rectus anterior quadriceps and vastus lateralis muscles were measured using tensiomyography. 

#### 2.5.8. Stiffness

It is a variable determined by the ratio between the force produced by the mechanical impulse and the depth of tissue deformation using myotonometry. Its unit of measurement is newtons per meter (N·m^−1^). In the present study, the rectus anterior quadriceps and vastus lateralis muscles were measured. To perform the measurement, the patient must be in a muscle relaxation position. The sensor of the device is then placed perpendicular to the most prominent part of the chosen muscle belly (only superficial musculature can be assessed). From this position, a small pressure is exerted with the device on the tissue so that it can perform three short pressure applications (15 ms) on the tissue [24]. Myotonometry was performed using the MyotonPro device (MytonPro, Myoton Ltds., Tallinn, Estonia).

### 2.6. Statistical Analysis

A statistical analysis was performed using SPSS v.26 statistical software. The level of significance was set at α = 0.05. The Shapiro-Wilk test was used to assess the normal distribution of the variables. Descriptive statistics were performed for all the variables analyzed. Quantitative variables were expressed by mean and standard deviation (in case of normal distribution) and median and interquartile range (for non-normal distribution). Qualitative or categorical variables were expressed as percentages. To evaluate the effect of the intervention, due to the sample size, a Wilcoxon test was performed using the pre- and post-intervention values.

## 3. Results

The baseline characteristics of the 16 participants are shown in Table 1. As shown, the majority of the sample (81.25%) were female, with a mean age of 76.5 (7.68) years. The sample had a BMI considered as "normal", and were independent in their daily lives. Moreover, as for the different tests of functionality, in all of them the mean was above the cut-off points. In terms of adherence, the average attendance was 83.5% (12.2). Table 2 shows the changes produced after the intervention both in the functional tests and in the variables related to functional quality.

## 4. Discussion

The present study aimed to evaluate the changes produced by an MCT program on both physical performance and muscle quality in a population of healthy older adults. The main results of this study show some moderate improvement in physical performance, but no effect in muscle quality parameters, except for vastus lateralis stiffness. 

As already explained in the introductory section, aging involves a series of local and systemic changes, which cause changes in physical performance as well as in muscle strength and quality. Specifically, Jacob et al. [25], concluded that older adults have lower physical performance, as well as longer contraction times and greater muscle stiffness.

In fact, loss of physical performance and muscle quality are key factors for the diagnosis of sarcopenia, and are in turn related to frailty and other adverse events, such as falls, loss of dependency and even mortality [26,27]. For its part, the EWGSOP recognizes that "there is no universal consensus on assessment methods for routine clinical practice" [6]. Fabiani et al. [28] propose tools such as tensiomyography as possible solutions to the assessment of muscle quality. Myotonometry is another current alternative in the assessment of muscle quality which provides complementary information to tensiomyography [29,30].

Regarding recommendations on the treatment of sarcopenia, there is a consensus that MCT with an emphasis on resistance training is one of the first-line treatment options [19]. Actually, given the high prevalence of sarcopenia among older adults, some authors have gone so far as to question whether the diagnosis of sarcopenia really matters, arguing that resistance training should be prescribed to that entire population group [31]. However, other authors recognize the particularity of this condition and the need for a specific therapeutic exercise prescription [32]. 

In view of the results in terms of muscle quality, the question would be whether myotonometry and tensiomyography are the best tools to assess it, or whether other tools such as electromyography, ultrasound or even blood biomarkers could provide us with better information on the state of skeletal muscle. Moreover, there is a need to study the effectiveness of novel training techniques such as blood flow restriction in terms of physical performance and muscle quality, as this could be a good alternative for older adults who cannot tolerate the high loads of conventional resistance training [33], or in combination with nutritional supplements [34].

This study has several limitations. First, the study has a relatively low sample size. In addition, the study lacks a control group, so the observed changes may have been caused by other factors, such as the tendency to decrease physical performance with aging. However, given that trends in physical performance and muscle quality tend to decrease with aging, it is unlikely that the improvements observed are due to other factors. As for the duration of the intervention, it is likely that a longer duration would have led to significant changes in muscle quality variables. Another limitation of the study is that the number of women was significantly higher than that of men. In this sense, it is possible that the results obtained in a population of men would be different. Finally, it is important to highlight that the population studied was older adults with a high level of independence. It is possible that in frail populations or those with comorbidities, the results obtained could be different.

## 5. Conclusions

MCT is able to generate improvements in the physical performance of older adults, but these improvements are not reflected in muscle quality parameters measured by tensiomyography and myotonometry. Intervention studies are needed with larger populations and control groups, as well as a lower level of functional capacity, in order to perform high intensity training.

## Figures and Tables

**Table 1 ijerph-20-00222-t001:** Baseline characteristics of the sample.

Variable	
N (M/F)	16 (3/13)
Age, years	76.5 ± 7.68
Height, cm	151 ± 5.01
Weight, kg	59.6 (10.3) *
BMI, kg/m2	27.6 ± 3.96
Barthel, score	100 (0.0) *
FES-I, score	18 (4.0) *
PASE, score	111 ± 41.3
Attendance, %	83.5 ± 12.2

*, Median (IQR); BMI, Body Mass Index; FES-I, Falls Efficacy Scale-International; PASE, Physical Activity.

**Table 2 ijerph-20-00222-t002:** Difference between baseline and post intervention.

Variable	Baseline	Mean Difference	95% Confidence Interval		Effect Size	*p*-Value
			Lower	Upper		
SPPB, score	12 ± 1.0 *	−1.00	−	−	1.00	0.346
5XSST, s	9.47 ± 1.75	1.50	0.10	2.71	0.72	0.027
TUG, s	9.56 ± 2.06	0.94	0.31	1.84	0.95	0.001
Walking Speed, m/s	1.05 ± 0.24	−0.06	−0.13	0.01	0.54	0.110
Handgrip Strength, kg	20.6 ± 7.9 *	−0.35	−2.70	2.33	0.05	0.893
Contraction Time-RA, ms	35.0 ± 7.15 *	2.37	−5.12	14.49	0.28	0.410
Maximal Radial Displacement-RF, mm	3.95 ± 1.95	−0.66	−2.53	0.30	0.51	0.126
Stiffness-RF, N·m^−1^	307 ± 53.2	7.89	−17.00	33.00	0.24	0.480
Contraction Time-VL, ms	1.77 ± 1.63 *	6.25	−4.15	22.73	0.33	0.327
Maximal Radial Displacement-VL, mm	26.4 ± 18.3 *	0.19	−0.91	1.02	0.15	0.666
Stiffness-VL, N·m^−1^	318 ± 35.4	29.00	11.00	50.50	0.94	0.007

*, Median (IQR), SPPB, Short Physical Performance Battery; 5XSST, Five Times Sit to Stand Test; TUG, Timed Up and Go test; WS, Walking Speed; HG, Handgrip Strength; Tc, Contraction Time; Dm, Maximal radial displacement; St, Stiffness; RF, Rectus Femoris; VL, Vastus Lateralis.

## Data Availability

According to MDPI Research Data Policies, the data presented in this study are available on request from the corresponding authors. The data are not publicly available due to privacy medical reasons.

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
