# Peer review of "Effectiveness of a Multicomponent Training Program on Physical Performance and Muscle Quality in Older Adults: A Quasi-Experimental Study"

_ijerph, 2022, doi:10.3390/ijerph20010222_

Round 1

Reviewer 1 Report

Effectiveness of a multicomponent training program on physical performance and muscle quality in older adults. A quasi-experimental study.

Labata-Lezaun and colleges describe a quasi-experimental study to assess the effectiveness of a multicomponent training (MCT) program in improving physical performance and muscle quality in healthy older adults, for which they perform several functional tests and measure muscle quality parameters. The conclusions –MCT improves physical performance but not muscle quality- is not surprising.

Moreover, they discuss the limitations of the study as low sample size and lack of control group. Size of sample group is in fact small, but moreover, it is biased, since there are 13 females and only 3 males. Under these circumstances it would be better to limit the study only to the female sex. As for the lack of a control group, what they discuss is rather controversial, the best control group is the same group prior to the study (which is analyzed). Duration of intervention is another raised issue. To this point, although it is probable that only 15 sessions would not be enough to change muscle size/composition, these sessions are only twice weekly and only 60 % attendance is required -although surpassed-, both of which seem low. 

On the other hand, the article is very well written and requires only minor style interventions.

Minor comments:

-          In addition, aging involves structural [2], metabolic changes [3], and neuroendocrine 31

changes [4]

-          In particular, multicomponent training (MCT) characterized by the combination OF at 54 least 3 types of training such as resistance, aerobic, balance, and/or flexibility training has 5

-          In fact, the latest international con- 57 sensus on exercise in the older adults promotes the implementation of MCT programs as 58 one the best strategy strategies for improving strength, balance and gait, as well as reducing falls 59 [19].

-          2.4.2. Four-meter walking test instead of 2.4.2.4

Reviewer 2 Report

This was a study to evaluate the effectiveness of MCT on physical performance and muscle quality in older adults. The study design was a single arm intervention with a pre- and post- assessments. While the study was well-conducted and well-explained, I have one major concern. It has been shown repeatedly that MCT improves physical performance in older adults. In this study, the improvements are small to non-existent, and the study design and sample size are limited, so that part of the study does not really add to the current knowledge. The main point of this study is that two new techniques for evaluating sarcopenia, tensiomyography and myotonometry, provide little information about muscle quality changes. In my opinion, this paper should focus more on the relevancy of the measurement techniques. Perhaps, if the stimulus were greater, and/or the participants were less able at baseline, then the new measurement techniques might have detected an improvement.

Minor changes:

Abstract.

·      Define MCT before using the acronym.

·      Modify the second sentence to emphasize that the novelty is that muscle performance and quality will be assessed simultaneously.

·      Explain the MCT just a bit so the reader gets a better idea.

Introduction

Line 50. Extrapolated to what?

Lines 58-62. There is a sentence about how much is known about MCT and older adults, and then the next sentence is how this is the first study about MCT for performance and quality. This needs to be teased out so the reader understands the significance. Again, I think that the main point of interest is the muscle quality measures.

Line 65. Remove this sentence. It seems like instructions.

Methods.

·      Exclusion criteria. I think A and G are the same

·      Variables. Please write these entire paragraph in full sentences. For example, line 125, “It is a test battery of 3 tests. Balance test…” This goes for all the 2.4 sections.

·      At the end of each section, the authors state something to effect of, ‘its reliability has been studied.’ Please state what the findings mean.

·      2.4.6 Contraction time. Line 156. “the time… the muscle takes to reach 10-90% of maximal radial displacement.” Please explain this a bit more to help the reader understand the underlying physiology of the measurement, not just the specifics of the technique sequence.

·      Statistical analysis. There are 13 pre-post variables by my measure, so there should be some sort of Bonferroni adjustment to the p-value to reduce type I error.

Results.

·      Is there a reason the proportion of women was so high? It would be helpful to explain in the discussion.

·      Tables 1 and 2 should be combined. The variables should be written out, otherwise the reader has to constantly refer to the table and back which is onerous.

Discussion

Lines 202-205. I think this should be reframed. There were six measures of muscle quality and only one showed significant effect, and even that one may have just been random effects due to many measures. IMHO this study found moderate effects on function and no effects on muscle quality

Lines 222-228. Not sure why the protein portion is relevant.

Lines 226-228. This should be later in the discussion when proposing other research

Conclusion. The first sentence is a great synopsis. I think the second sentence should call for high intensity/volume interventions with lower level participants, which might show some muscle quality changes.

Author Response

Please see the attachment. Reviewer 2

Round 2

Reviewer 2 Report

The authors have address all of my feedback.